# *OTX* Genes in Adult Tissues

**DOI:** 10.3390/ijms242316962

**Published:** 2023-11-30

**Authors:** Alessandro Terrinoni, Giovanni Micheloni, Vittoria Moretti, Sabrina Caporali, Sergio Bernardini, Marilena Minieri, Massimo Pieri, Cristina Giaroni, Francesco Acquati, Lucy Costantino, Fulvio Ferrara, Roberto Valli, Giovanni Porta

**Affiliations:** 1Department of Experimental Medicine, University of Rome Tor Vergata, Via Montpellier 1, 00133 Rome, Italy; 2Genomic Medicine Research Center, Department of Medicine and Surgery, University of Insubria, Via JH Dunant 5, 21100 Varese, Italy; 3Department of Industrial Engineering, University of Rome Tor Vergata, Via del Politecnico 1, 00133 Rome, Italy; 4Department of Medicina e Innovazione Tecnologica, University of Insubria, Via JH Dunant 5, 21100 Varese, Italy; 5Department of Biotechnology and Life Science, University of Insubria, Via JH Dunant 3, 21100 Varese, Italy; 6Department of Molecular Genetics, Centro Diagnostico Italiano, Via Saint Bon 20, 20147 Milano, Italy

**Keywords:** *OTX* homeobox genes, adult tissues, tumor, inflammation, ischemia

## Abstract

*OTX* homeobox genes have been extensively studied for their role in development, especially in neuroectoderm formation. Recently, their expression has also been reported in adult physiological and pathological tissues, including retina, mammary and pituitary glands, sinonasal mucosa, in several types of cancer, and in response to inflammatory, ischemic, and hypoxic stimuli. Reactivation of *OTX* genes in adult tissues supports the notion of the evolutionary amplification of functions of genes by varying their temporal expression, with the selection of homeobox genes from the “toolbox” to drive or contribute to different processes at different stages of life. OTX involvement in pathologies points toward these genes as potential diagnostic and/or prognostic markers as well as possible therapeutic targets.

## 1. Introduction

Orthodenticle homeobox protein 1 and 2 (OTX1 and OTX2) are transcription factors encoded by homeobox-containing genes, embedded in humans on chromosomes 2p13 and 14q21-22, respectively [1]. Protein structures and interactome are described in Figure 1, and the list of interactors of OTX proteins is available in Appendix A.

*OTX* genes in vertebrates and *OTX-like* genes in lower species (e.g., the Drosophila orthodenticle *otd* gene) are indispensable for the specification, regionalization, and terminal differentiation of the rostral part of the central nervous system [2], involved in the development and morphogenesis of the neuroectoderm and the vertebrate central nervous system. During embryonic development, they are also important for specification of cell identity, cell differentiation, and the positioning of the body axis [3]. Their pleiotropic activities thus involve OTX proteins in the morphogenesis and physiology of different tissues and districts. They can each act singularly on specific cells and tissues or cooperate together in common pathways. Moreover, the different OTX proteins can coopt specific or shared effectors in their action (Figure 2). The main loci of their action are summarized in Figure 2 and specifically discussed in the text.

Mutations and abnormal expression of these powerful transcription factor genes are also involved in several human pathologies. This review aims to highlight the activity of *OTX* genes in postnatal tissues, mainly focusing on pathological conditions summarized in Figure 3 and discussed throughout the text.

## 2. Development

Analysis of *OTX1* and *OTX2* in human fetal brains revealed spatiotemporal distribution of mRNA and proteins during development [4]. *OTX2* is expressed in the diencephalon, mesencephalon, and archicortex, whereas *OTX1* can be considered a marker of proliferative zones of the neocortex in fetal brain development [4].

Double deletion of *Otx1* and *Otx2* in mice is embryonically lethal [5], whereas heterozygous double mutants show several defects in central nervous system and sensory organ formation due to gastrulation impairment [6,7].

*Otx1* knockout (KO) in mice leads to spontaneous epilepsy and seizures [8], and to repression of the differentiation, but not proliferation, of neuronal progenitor cells (NPCs), suggesting a role in controlling the differentiation–proliferation balance [9,10]. Conditional KO in developing neocortex led to its reduction in size and cell number [11].

*Otx2* KO revealed a peculiar idiosyncratic role in developmental processes involving septum formation and neocortex specification, and also in neurogenesis, oligodendrogenesis, and the regulation of cholinergic neurons in median ganglion eminences [12,13]. *Otx2* deletion in the thalamus instead shifted NPC differentiation from glutamatergic to GABAergic interneurons [14].

*Otx* genes are also expressed in sensory organs [6,8,15]. *Otx1* and *Otx2* play an essential role in proper mouse retina development in a dose-dependent fashion [16,17,18]. In human fetal retina *OTX2* is expressed first in the dorsal portion of the optic vesicles and maintained in the outer layer of optic cup, from which the retinal pigmented epithelium (RPE) originates [19]. In mice it also induces the production of pigment in RPE through the induction of genes involved in melanosome glycoprotein formation [17]. Later, it is also expressed in part of the neural retina (NR), particularly in post-mitotic neuroblast cells that generate different cell types including ganglion cells, photoreceptors, glial, and Müller cells [20]. *OTX1* expression is instead confined to the anterior retina [19]. The importance of these two genes is reinforced by the fact that loss-of-function mutations in *OTX1* and *OTX2* lead to variably severe ocular malformations such as Microphthalmia-anophthalmia-coloboma (MAC) [21,22,23,24].

In addition to its role in eye formation, *OTX* genes are also involved in inner ear development in mammals. In mice, *Otx1* and *Otx2* are expressed in non-sensory regions, presumptive lateral crista (*Otx1*) and ventrolateral part (both genes), whereas mutations in both genes are associated with utricle, saccule, and cochlea developmental defects [25]. OTX2 regulates the expression of TAp63, a crucial factor in proper inner ear formation [26].

*Otx2* expression is strictly correlated with expression of *N-myc*, another gene involved in ear development. *Otx2* is expressed in the roof of the cochlear duct, and its inactivation in mice leads to the shift of normally expressing *Otx2* regions from non-sensory to prosensory, marked by the lack of formation of the Reissner’s membrane, the formation of two organs of Corti, and the dysregulated proliferation of hair cells in the apical portion of the cochlea [27]. Furthermore, several defects are evident in the macula and saccule of *Otx1*^−/−^ mice, and markedly worse in *Otx2*^−/+^ *Otx1*^−/−^ animals. Interestingly, the defect in *Otx1*^−/−^ can be rescued by overexpressing an *Otx2* cDNA, demonstrating a fundamental but overlapping role of both genes in driving the correct architecture of the organ [25].

## 3. Embryonic and Adult Stem Cells

In addition to what has already been described earlier, in early embryonic development, OTX2 plays a critical role in maintaining embryonic stem cells (ESCs) in a metastable state, in order to fluctuate between different states of pluripotency. In particular, OTX2 is required to predispose ESC for differentiation and the morphogenetic process of gastrulation [28].

In pigmented epithelium transplantations approaches in retina degeneration, the limited clinical benefit is mainly due to the dedifferentiation of the transplanted cells that undergo an epithelial–mesenchymal transition. In adult tissue, the role of the homeogene OTX2 in preventing dedifferentiation through the regulation of target genes has been demonstrated: OTX2 transfected photoreceptors transplanted in a damaged retina prevented the epithelial–mesenchymal transition [29].

## 4. Adult Tissues

*OTX* genes are involved in physiological functions in some adult tissues as well (Figure 2).

### 4.1. Neuronal Tissues

#### 4.1.1. Brain

In adult mouse brain, *Otx2* is expressed in several regions, including the ventral segmental area (VTA), lateral geniculate nucleus, superior colliculus, medial septum, the cerebellum, and choroid plexus [30]. The choroid plexus, located in brain ventricles, is responsible for the synthesis of cerebrospinal fluid (CSF), where epithelial cells secrete OTX2 [31,32,33]. This provides the protein to all of the cortex, where OTX2 exerts a non-cell autonomous activity, for example, in the supragranular layer of the binocular visual cortex and in anxiety-related regulation of behavior [34,35,36].

OTX2 regulates critical periods (CPs) of plasticity in adult brain cortex, where environmental stimuli induce a learning process that deeply affects neuronal physiology and morphology. Since the 1960s, the most studied process involves the visual cortex, where OTX2 is involved in a positive feedback loop with perineuronal net (PNN) in the induction and termination of CPs [30,37,38]. In detail, in response to an external stimulus, i.e., a sensory input, PNN starts to form in the visual cortex. Glycosaminoglycans (GAGs) contained in this specialized extracellular matrix bind OTX2 [37], promoting its internalization by GABAergic inhibitory interneurons that synthesize parvalbumin (PV-cells). OTX2 in PV-cells mediate their maturation, which is necessary and sufficient to initiate CP [30,32,34]. The accumulation of OTX2 in PV-cells in mice increases from P20 to P40 when CP ends. OTX2 effect follows a ‘French flag’ temporal model, characterized by two thresholds: the first that initiates CP characterized by OTX2 accumulation in PV-cells, and the second that causes CP closure and non-plastic state maintenance in which OTX2 concentration does not change [37,38,39].

Transient inactivation of OTX2 via delivered agents (i.e., siRNA) leads to the reactivation of plasticity in adult visual cortex. Thus, the differential kinetics of OTX2 regulation across brain regions might offer opportunities for therapeutic intervention in neurodevelopmental disorders [32,40].

OTX2 produced by the choroid plexus also exerts a non-cell-autonomous role in support cells, i.e., astrocytes, in ventricular-subventricular zones (V-SVZ) and rostral migratory stream (RMS) in adult brain via the regulation of extracellular matrix (ECM) composition. This effect is essential for both newborn neuron levels and olfactory bulb formation [41,42].

Brain plasticity associated with OTX2 also extends to the primary auditory and medial prefrontal cortex, associated with hearing and acoustic effects [43].

#### 4.1.2. Dopaminergic Neurons

*Otx2* expression is essential for proper dopaminergic (DA) neuron development and maintenance [44,45,46]. In particular, the depletion of *Otx2* resulted in a severe reduction in DA neurons, especially those located in the VTA; abnormal innervation of the mesolimb, A10 DA axonal projection; and loss and alteration of sensitivity to drugs [47,48], whereas its upregulation resulted in increased innervation, resistance to neurotoxins, and reduction in spontaneous locomotor activity [49]. Detailed analysis of OTX2 action in adult mesencephalic–diencephalic DA neurons has been carried out by Simeone and colleagues [50].

#### 4.1.3. Retina and PVR

*Otx* expression is maintained in adult vertebrate retina, especially in the RPE [15,16,20], and the OTX2 protein has been observed in some retinal cells [51,52]. The analysis of the expression of a subset of genes in retinectomy samples from patients affected by proliferative vitreoretinopathy (PVR), an inflammatory complication of retinal detachment, showed an association between *OTX* genes expression and severity of the disease: those samples expressing *OTX2, VEGFA, TP53,* and *TP63* are characterized by more severe PVR and patients require a greater number of surgical procedures, whereas samples with high *OTX1* expression came from patients with a better prognosis [51].

The presence of OTX2 in differentiated cells of the retina is compatible with its role in cell identity maintenance, while its upregulation in PVR samples can be associated to the reactivation of proliferation in RPE cells released in the vitreous humor as a consequence of retinal detachment [53].

*OTX1* expression in MIO-M1, a Müller cell line, is altered in response to a hypoxia-mimicking treatment with cobalt chloride. Interestingly, this gene is upregulated in the recovery phase after cobalt chloride has been removed, suggesting a differentiating role in processes activated by hypoxic stimulus in glial cells [54].

#### 4.1.4. Pineal Gland

In addition to *Otx* action in pineal gland formation [18,55], OTX2 is also present in melatonin-producing pinealocytes, where it controls the expression of Tph1, Aanat, and Asmt—enzymes responsible for melatonin synthesis [56].

#### 4.1.5. Pituitary Gland

OTX2 is also involved in pituitary gland development through the control of HESX1 and POU1F1 transcription factors [57,58], and in adult functioning, as observed in patients where deletion or mutations of *OTX2* are associated with pathologies with variable expression, ranging from anophthalmia, ear abnormalities, and hypopituitarism [59] to pituitary hypoplasia and defects in pituitary hormone production [58,60,61,62,63].

*Otx2* is also expressed in mice hypothalamus, where it induces the gonadotropin-releasing hormone (GnRH) [64,65].

#### 4.1.6. Sinonasal Mucosae and Nasal Polyps

Finally, *OTX* genes are also involved in pathological conditions linked to inflammation. They are expressed both in normal sinonasal mucosae and in nasal polyps (NPs) [66], inflammatory outgrowths of sinonasal tissue, usually presenting as bilateral inflammatory lesions originating in the ethmoid sinuses and projecting into the nasal airway beneath the middle turbinate [67]. In NPs the number of p63-positive cells in the epithelium increases [68,69], with the ratio between TAp63 and ΔNp63 isoforms correlating with polyp recurrence [66], consistent with a potential p63 pro-proliferative or oncogenic function [70,71].

In this setting, OTX2 is crucial in determining the progression and possibility of recurrence, due to its ability to transactivate TAp63, moving cells toward a differentiated, un-proliferative state associated with lower probability of recurrence compared to cases in which TAp63 is less expressed [66].

### 4.2. Breast

Homeobox genes are well studied in mammary gland development, due to their ability to direct transitions necessary to switch between linear and cyclical phases [72,73], but little is known on the involvement of *OTX* genes in this process.

*OTX1* is physiologically expressed in breast tissues during the linear and cyclical organ phases, with a role in cell differentiation and the balance between symmetrical and asymmetrical division, and it is overexpressed during lactation in mice [74,75].

### 4.3. Hematopoiesis

*OTX1* expression is also involved in the control of blood cell production. *Otx1* gene is transcriptionally active at several hematopoietic sites, i.e., in bone marrow, and especially in erythroid lineage cells [76]. Indeed, *Otx1* deficient mice show fewer red blood cells and a reduction in early and late erythroid progenitors; furthermore, they show aberrant numbers of leukocytes and myelo-monocytic precursors [76]. In a cellular model of Shwachman–Diamond Syndrome (SDS), a rare ribosomopathy characterized by altered hematopoiesis [77], *OTX2* has been found to be downregulated [78].

### 4.4. Myenteric Plexus/Intestine

#### 4.4.1. Inflammation in Mice

Upregulation of *Otx* genes has also been described in the myenteric plexus along the gastrointestinal tract in response to an inflammatory challenge. Bistoletti et al. demonstrated that in response to dinitro-benzene sulfonic (DNBS) acid-induced colitis, there is a significant increase in both *Otx1* and *Otx2* mRNA and corresponding protein levels in longitudinal muscle myenteric plexus (LMMP) preparations of rat small intestine and distal colon. Anti-OTX antibodies showed an increase in the number of myenteric neurons expressing the transcription factors in distal colon and in the small intestine—i.e., far from the site of injury—suggesting that these molecules can also be induced at distant sites [79]. It is possible that distant OTX1 and OTX2 upregulation is caused by inflammatory mediators such as VEGFα, positively correlated with *OTX2* expression in a report examining retinal pigment epithelial cells during inflammation [51], or TNFα, indicated as a modulator of *OTX2* expression in in vitro models of chronic subretinal inflammation [80]. In addition, a positive correlation between inflammatory cytokines (i.e., IL6) and *OTX1* has been seen in a genome-wide study of foot-and-mouth viral disease in animals [81]. We can thus suggest that *OTX* genes are implicated in neuronal degeneration during inflammatory states along the gastrointestinal tract, suggesting *OTX* genes as potential targets for the development of new therapeutic approaches [82].

#### 4.4.2. Inflammation in Zebrafish

Likewise, the induction of inflammation in adult zebrafish intestine through a soy-based diet showed an increase in *otx1* and *otx2* expression that parallels gut morphological alterations in the acute phase of inflammation, again suggesting a role in remodelling processes in response to inflammatory stimuli [83].

#### 4.4.3. Ischemia/Reperfusion in Mice

OTX proteins are also upregulated during intestinal ischemia/reperfusion (I/R) injury and are correlated with alterations of the intestinal neuromuscular function in this pathophysiological condition (Figure 4 shows characteristic immunohistochemistry of the specific localization).

Filpa et al. demonstrated that nitric oxide (NO) is involved in OTX1 and OTX2 ischemic-induced upregulation, describing an interplay between both transcription factors and enteric nitrergic neurons that results in altered motor responses involving NO production [84]. During both gut inflammation and I/R injury, inducible nitric oxide synthase (iNOS) and neuronal nitric oxide synthase (nNOS) exert neurodamaging and neuroprotective actions, respectively, in enteric neuronal homeostasis [84,85]. NO derived from iNOS promoted OTX1 up-regulation predominantly in enteric glial cells and in few myenteric neurons; whereas nNOS is more closely related to OTX2 up-regulation, which is only seen in the soma of a relatively small percentage of myenteric neurons, sustaining the protective role of OTX2 already noted in NPs [66,84]. The linkage of OTX1 and OTX2 and NO pathways to cell damage and inflammatory activity suggests that OTX transcription factors can be interesting targets for the treatment of gastrointestinal problems such as I/R injury [86].

### 4.5. Mastocytosis

A genome-wide association (GWAS) in patients affected by mastocytosis revealed a potential novel involvement of *OTX2* [87]. Mastocytosis comprises a heterogeneous group of diseases which causes abnormal accumulation of clonal mast cells (MC) in skin, bone marrow, and/or other visceral organs [88]. SNP-array analysis detected polymorphisms associated with the disease, including polymorphism rs11845537 G > A in the *OTX2-AS1* gene, which is frequently observed both in adults and in children with cutaneous mastocytosis [87]. *OTX2-AS1* belongs to the family of natural antisense transcripts (NATs), a class of RNAs having sequence complementarity with other transcripts and involved in several cellular processes ranging from proliferation to EMT and tumorigenesis [89]. Its expression has also been observed in developing retina [90]. However, a potential role of the *OTX2-AS1* gene and the subsequent involvement of *OTX2* in mastocytosis is still undetermined [87].

## 5. Tumor Progression

### 5.1. Medulloblastoma

The main tumor in which *OTX* genes have been studied is medulloblastoma (MB), the most common brain tumor in children, with virtually all medulloblastomas expressing *OTX1, OTX2,* or both genes [91].

MB originates in the cerebellum, where OTX2 is detectable during development in progenitor cells of the external granule layer [92] and then becomes restricted to choroid plexus, pineal gland, and retinal pigmented epithelium postnatally [93].

MB is divided into five molecular subgroups (WNT, SHH/wild type TP53, SHH/mutant TP53, Group 3 and Group 4) with unique genetic, epigenetic, and molecular signatures [94,95]. *OTX2* is highly expressed in Group 3 and Group 4, and is expressed in WNT MB but not in SHH, which usually shows *OTX1* expression [95,96]. The expression of these genes is also associated with the localization and prognosis of these tumors: *OTX1* is expressed in a high proportion (53%) of MB, especially in nodular/desmoplastic tumors, normally localized in the hemisphere and large cell variants, and it is associated with either very young or adult age of onset. *OTX2* is also expressed in a large proportion (>66%) of medulloblastomas and is specifically associated with vermian topography and classic, large cell, and anaplastic variants. This gene was also associated with the development of leptomeningeal metastasis and shorter overall survival [91,97,98].

Various studies have demonstrated that all-trans retinoic acid (ATRA) repressed *OTX2* expression and inhibited *OTX2*-expressing medulloblastoma cell growth, suggesting that medulloblastomas may be amenable to therapy with retinoids [99]. In fact, pharmacologically relevant doses of ATRA induce apoptosis in medulloblastoma cells—although no connection with anaplastic histology or inhibition of *OTX2* expression was established [100,101].

In addition to its expression levels, *OTX2* is also duplicated in several MB with no WNT or SHH activated pathways [102], thus leading several studies to elucidate an *OTX2* oncogenic role in medulloblastomas.

In *OTX2* expressing MB cell lines, silencing of the gene caused downregulation of directly targeted cell cycle genes and indirectly targeted genes for visual perception, and the induction of differentiation to a neuronal-like status, recalling events in cerebellum development [103]. Surprisingly, overexpression of this gene in MB cell lines lacking endogenous *OTX2* expression initially induces cell cycle progression but finally inhibits cell proliferation, activating a senescence-like phenotype that involves the P53 pathway and/or secretion of senescence-associated factors, suggesting that OTX2 is unable on its own to control cell cycle [96].

In fact, oncogenic properties exerted by OTX2 are associated with MYC, another typically amplified and/or overexpressed gene in medulloblastoma [104], which is directly induced by OTX2 [96] and frequently colocalizes with OTX2 [105] to promoters of MB [104] and embryonic/neural stem cell specific genes [106].

As described for other transcription factors like CRX, it is possible that OTX2 exerts a non-canonical transcription factor activity that involves histone acetylation and methylation of specific promoters [107]. This hypothesis suggests that OTX2 interacts with ATXN7, promoting histone acetylation through recruitment of histone acetyl transferase (HAT)-containing activators, even if direct evidence of this mechanism is lacking [105]. In addition, Group 3 and 4 MB have high levels of trimethylated histone 3 lysine 27 (H3K27me3) [108] either in the presence or absence of mutation in H3K27 demethylases [108], and *OTX2* silencing in MB cell lines resulted in downregulation of polycomb genes, that are required for H3K27 methylation, and upregulation of H3K27 demethylases, causing a decreased in trimethylated H3K27 especially in OTX2-binding promoters [109]. One effect of this chromatin landscape regulation is the downregulation of axon guidance signaling genes in Group 3 and Group 4 MB, and specifically of semaphorin ligand and receptors, which guides differentiation in embryonic neurons [95]. Further studies in Group 3 MB revealed a OTX2-PAX3 signaling axis that controls cell fate [110].

Taken together, these data indicate that OTX2 action in MB varies in different MB subgroups [110].

### 5.2. Retinoblastoma

RB is the most common ocular cancer, and it often reveals morphological features suggesting its origin from photoreceptor cells (e.g., formation of fleurettes and Flexner–Wintersteiner rosettes) [111]. This tumor mainly occurs due to biallelic mutation of the *RB1* gene on chromosome 13; but other events, such as chromosomal anomalies (isochromosome 6p or extra copies of chromosome 1p) [111] or epigenetic changes [112], are involved in RB development. RB protein is also involved in several cancers due to its ability to interact with p53 and p21 and regulate cell cycle and apoptosis [113].

Due to the toxicity of standard treatments [114,115], several efforts have been made to identify other candidate therapeutic targets [112], with a strong interest in genes related to stemness due to the peculiar role of cancer stem cells in tumor development, treatment, and response to therapies [116,117].

Analysis of expression profiles of RB tumor samples and cell lines identified several genes and proteins overexpressed in this pathology [52,118]. Interestingly, all these genes are associated with *OTX2*, either directly or indirectly. The high expression of *OTX2* and *CRX* in RB tumors and cell lines suggested that retinoblastomas may originate from cells normally expressing these transcription factors, such as bipolar cells or photoreceptor precursors [52]. Expression of HIWI2, a PIWI-like protein involved in stem cell self-renewal, is increased in Y79 cells, and its knockdown significantly downregulates OTX2 protein and gene expression, perhaps through PI3K/Akt or FGF signaling pathways [119,120].

We recall that CRX and OTX2 have important roles in the development of human retina: CRX is involved in the proliferation of cells and in cells committed to the bipolar lineage, whereas OTX2 is associated with the maturation of photoreceptor cells. In mature retina, these genes show a reversed expression pattern: CRX is most abundant in photoreceptors and OTX2 is primarily expressed in bipolar cells [121].

OTX2′s crucial role in cell fate determination in mouse and human retina [52,122] pointed toward this gene as a potential target for new therapies for RB. Effectively, *OTX2* knockdown (KD) by siRNA or downregulation through treatment with ATRA resulted in less proliferation and increased cell death in cell lines and tumor growth reduction in vivo, suggesting that OTX2 could indeed link multiple tumor-driving pathways involving *CRX*, *C-MYC* and RB phosphorylation [123].

### 5.3. Sinonasal Neoplasms

Analysis of the expression of *OTX* genes in tissues of the nasal cavity revealed a significant modulation in neoplastic tissue, suggesting that the activation/inactivation of *OTX* genes is involved in the pathogenesis of different types of sinonasal neoplasms [124,125]. Analysis by real time PCR and immunohistochemistry revealed that both genes are expressed in normal sinonasal mucosa. Furthermore, high expression of *OTX1* was detected in non-intestinal-type adenocarcinomas (NITACs), whereas *OTX2* was present in olfactory neuroblastomas (ONs) and poorly differentiated neuroendocrine carcinomas (PDNECs). Interestingly, neither *OTX1* nor *OTX2* was detected in intestinal-type adenocarcinomas (ITACs) [124,125]. Intriguingly, it has been shown that chromosomes 2 and 14, on which *OTX1* and *OTX2* genes map, are present in extra copies in ON samples of many patients, suggesting a mechanism associated with chromosomal dosage [126].

Taken together, these data point toward *OTX* genes and protein as markers for proper tumor classification and potential new targets for therapy.

### 5.4. Laryngeal Squamous Cell Carcinoma

*OTX1* has an oncogenic role in laryngeal squamous cell carcinoma (LSCC) tumorigenesis and progression. LSCC is one of the most common cancers occurring in the head and neck region [127,128]. Tu and colleagues demonstrated for the first time that the overexpression of *OTX1* in LSCC tumor samples is associated with lymph node metastasis and poor prognosis. *OTX1* KD resulted in reduction of proliferation, migration, and invasion capacity in LSCC cell lines and diminished growth in xenograft. They also found that *OTX1* is negatively regulated by miR-129-5p [129]. This miRNA has already been studied in many types of cancers [130,131,132] and microRNAs (miRNAs) are strongly associated with the development and lymph node metastasis of laryngeal cancers [133,134]. Therefore, the upregulation of miR-1295p could be a potential therapeutic strategy for patients with LSCC and high *OTX1* expression [129].

### 5.5. Esophageal Squamous Cell Carcinoma

Esophageal squamous cell carcinoma (ESCC) is a multifactorial disease characterized by a low survival rate due to late-stage diagnosis, with a high incidence in both Asian and Western countries [135,136]. Comparative analysis of ESCC and adjacent noncancerous tissues revealed that an increased expression of *OTX1* in ESCC is associated with tumor size, lymph node metastases, and survival [136]. Studies of overexpression and silencing in cell lines indicate that OTX1 promotes migration and invasiveness in vitro and tumorigenesis in nude mice xenograft models [136].

### 5.6. Gastric Cancer

Gastric cancer (GC) tissues also present a high expression of *OTX1* compared with adjacent non-tumor tissues, both at the mRNA and protein levels, [137,138] with higher levels in GC that develops lymph node metastasis and in patients with lower survival rate [138].

*OTX1* KD in GC cell lines caused a reduction in proliferation, with cell cycle arrest in the G0/G1 phase and less migration and invasion via the reduction of the expression of mesenchymal markers and EMT-related transcription factors, a critical step involved in cancer metastases [139]. Furthermore, apoptosis is increased in response to *OTX1* KO [138].

Analogously to what is observed in other neoplasms, OTX1 activity is associated with levels of miR-3196, a GC commonly downregulated microRNA, with the *OTX1* 3′-UTR (untranslated region) as a target [140].

These findings demonstrate an OTX1 role in GC carcinogenesis, promoting the metastasis of GC cells by the induction of the EMT process [138].

### 5.7. Colon/Colorectal Cancer

*OTX1* overexpression is also implicated in colorectal cancer (CRC) development and progression. This gene is commonly overexpressed in CRC tissues and leads to tumor growth in vivo and cell proliferation and invasion in vitro. Inhibition of *OTX1* expression instead reduces proliferation and invasiveness in vitro [141].

It was further demonstrated that induced upregulation of *OTX1* in CRC cell lines causes an epithelial–mesenchymal transition (EMT)-like phenotype in CRC cells, as shown by the up-regulation of mesenchymal markers (N-cadherin and Vimentin) and EMT related transcription factors (i.e., Twist1, Snail, Slug, and ZEB1) and the concomitant down-regulation of epithelial marker E-cadherin [141].

EMT regulation in CRC also involves other molecules, such as long noncoding RNA (lncRNA) FEZF1-AS1, which is overexpressed in CRC tissues and cell lines and positively affects OTX1 protein levels [142]. Furthermore, its inhibition reduces EMT activation via the downregulation of OTX1 protein [142].

Similarly, *OTX1* interacts with lncRNA HNF1A-AS1 and PBX3 in colon cancer to activate the extracellular-signal-regulated kinase/mitogen-activated protein kinase (ERK/MAPK) pathway and promote angiogenesis via the PBX-OTX1-VEGF axis [143].

Therefore, OTX1 control of EMT and other pathways could be a potential target for the therapy of colon cancer [141].

### 5.8. Hepatocellular Carcinoma

OTX1 further contributes to Hepatocellular carcinoma (HCC) progression by regulating the ERK/MAPK pathway. The expression level of *OTX1* was significantly elevated in HCC tissue compared to paired non-cancerous controls, and *OTX1* silencing resulted in cell growth retardation, cell cycle arrest in the S phase, and decreased phosphorylation of ERK/MAPK signaling [144,145]. *OTX1* expression was also positively correlated with the expression of a lncRNA, MAFG-AS1, whose inhibition suppressed proliferation, migration, invasion, and angiogenesis in HCC [146].

### 5.9. Pancreatic Cancer

Pancreatic cancer (PC) is one of the most lethal tumors, ranking as the 7th leading cause of global cancer deaths in the developed world [147]. Studies revealed that *OTX1* is highly expressed in pancreatic cancer tissues and cell lines [148,149], and that it interacts with multiple factors in PC.

OTX1 is a downstream target of miR-4516, which is down-regulated in pancreatic cancer tissues and cell lines, suggesting its role as a tumor suppressor. Its overexpression inhibited pancreatic cancer cell proliferation, migration, and invasion, via negatively regulating OTX1 [148], pointing toward miR-4516/OTX1 interaction as a novel therapeutic target for PC. Moreover, miR-4269 overexpression inhibits pancreatic cancer cell proliferation, migration, and invasion by affecting the E-box binding homeobox 1 (ZEB1)/OTX1 pathway [149]. ZEB1 is a zinc-finger-homeodomain transcription factor that regulates cell growth and differentiation [150,151]. Bioinformatic prediction revealed that ZEB1 could bind to *OTX1* promoter and activate its transcription [149]. MiR-4269, modulating ZEB1/OTX1 axis, exerts tumor inhibitory effects on pancreatic cancer progression and offers a new insight into the clinical treatment of pancreatic cancer patients [149].

### 5.10. Breast Cancer

*OTX1* is overexpressed in breast cancer samples, and its expression correlates with p53 levels. Analysis of interaction between the two revealed that p53 directly binds the *OTX1* promoter, inducing its expression [74].

Analysis of OTX1 in LA7, a breast cancer stem cell (CSC) line, demonstrated that p53 and OTX1 levels increased when these cells were stimulated to differentiate, suggesting their involvement in asymmetrical division of CSCs [74]. These data point toward OTX1, together with p53, as a central molecule in the breast cancer stem cell symmetric/asymmetric division balance, similar to what observed in mammary SCs [75,152].

Similar to observations in other tumors, lncRNA and miRNA modulate *OTX1* gene expression. In particular, miR-3196, which negatively regulates OTX1, is “sponged” by ADPGK-AS1 lncRNA, whose overexpression in breast cancer is a potential prognostic factor [153]. The depletion of miR-3196 allows OTX1 to exert its proliferating effect [153].

### 5.11. Lung

Non-small cell lung cancers (NSCLC) account for approximately 85% of lung cancers [147] and can be subdivided into lung adenocarcinoma (LUAD), lung squamous cell carcinoma (LUSC), and large-cell carcinoma (LCC).

In LUAD, DMBX1 (diencephalon/mesencephalon homeobox 1), a homeodomain-containing transcription factor of the bicoid sub-family, interacts with OTX2, preventing it from directly activating p21 transcription, thus causing cell cycle blockade in G1/S [154], an event already described in MB [103].

Analysis of NSCLC tissues and cell lines revealed an overexpression of *OTX1* and its silencing revealed a central role in promoting proliferation, migration, and malignant progression in this pathology [155]

### 5.12. Bladder Cancer

There are more controversies surrounding OTX1 involvement in bladder cancer, which includes several types of cancer arising from bladder and upper urinary tissues and arises when bladder epithelial cells become malignant [156]. A genome-wide methylation analysis on bladder cancer and healthy bladder tissues identified, among others, *OTX1* as a tumor-specific highly methylated gene [157]. Starting from these data, Beukers et al. identified OTX1, FGFR3, and TERT as a combined diagnostic urinary biomarker in samples of patients affected by primary non-muscle invasive bladder cancer (NMIBC), either for detection or recurrence in high-grade primary tumors [158].

However, a more recent study on bladder cancer and healthy tissues and cell lines detected the overexpression of *OTX1* in cancer that correlates with poor prognosis in patients. KD and overexpression experiments in cell lines identify an association between *OTX1* expression and motility, cell cycle progression and tumor growth in xenografts [159].

The analysis of exosomes from urine of patients affected by high-grade muscle-invasive bladder cancer identifies the presence of *OTX2-AS1,* but functional studies on *OTX2* involvement remain to be carried out to clarify its possible involvement in pathology [160].

### 5.13. Lymphoma

*OTX* genes are also involved in lymphomas. *OTX1* is overexpressed in Hodgkin’s lymphoma (HL) cell lines, whereas *OTX2* and *OTX2-AS1* have been detected only in the KM-H2 line [161]. Analysis of primary patient samples revealed that both *OTX1* and *OTX2* are overexpressed, while *OTX2-AS1* levels remains analogous to those observed in B-cells, suggesting that it does not have a role in HL [161]. Furthermore, multiple copies of these genes are present in HL cell lines [161]. *OTX2* expression in KM-H2 line is induced by aberrant FGF2-pathway signaling, with OTX2 controlling *MSX1* and *FOXC1* transcription factor expression.

Interestingly, the typical T-cell, zinc-finger, and homeobox gene ZHX1 is upregulated by both *OTX* genes and may sustain the deregulated differentiation of B-cells in HL [161].

*OTX1*, but not *OTX2*, is expressed in specific subsets of B-cell non-Hodgkin’s lymphomas (NHL): in particular, Omodei and colleagues detected *OTX1* expression in nearly all diffuse large B-cell lymphomas, Burkitt lymphomas, and high-grade follicular lymphomas, but not in precursor-B lymphoblastic lymphoma, chronic lymphocytic leukemia, marginal zone, and mantle cell lymphomas, or multiple myeloma. A subset of germinal center (GC) B-cells carrying plasma markers also express *OTX1*, suggesting a role in B-cell differentiation [162].

These findings hint that *OTX1* levels might be useful as a molecular marker for high-grade GC-derived NHL and its involvement in B-cell lymphomagenesis [162].

## 6. Conclusions

In this review, we examined studies that implicate the functions of *OTX* genes in a variety of pathological conditions as well as normal differentiation processes. Interestingly, the sometimes unique and sometimes joint action of OTX1 and OTX2 reflects the evolutionary amplification of the function of genes by varying the time and place of their expression.

The best example of this process is represented by the mammary gland, where homeobox (HB) genes have a critical role in both cell growth and differentiation. In normal mammary gland, HB genes are involved in ductal formation, epithelial branching, and lobulo–alveolar development by regulating cell proliferation and differentiation [73]. HB genes are controlled in a spatial and temporal manner in both stromal and epithelial cells and when homeobox genes are misexpressed in animal models, different defects are displayed in mammary gland development. During the cyclical development of the mammary gland, the *OTX1* gene is overexpressed in lactation, confirming a role of this transcription factor in cell differentiation [75]. Additional data show different *OTX1* gene expression levels in mice breast tissues during the linear and the cyclical organ phases, suggesting its function in normal stem cell differentiation [74].

In adulthood, *OTX1* and *OTX2* expression is maintained in only limited tissues, most of which are reasonably associated with nervous system, such as choroid plexus, dopaminergic neurons, enteric nervous system, pituitary and pineal glands, or sensory organs such as retina or sinonasal mucosae, but also with non-neuronal tissues such as breast or hematopoietic sites.

Their frequent expression in pathological conditions associated with inflammation, as described in retinae, nasal polyps, intestines, or cancer shed a new light on these transcription factors as potential drivers of pathologies, and thus potential therapeutic targets.

The behavior of these genes strengthens the hypothesis that homeobox genes from the “toolbox” can be spatially and temporally selected for their function during the entire lifespan, from embryonic development to adult life [163]. Of particular interest are the findings that the *OTX* genes have important roles in tumor development and inflammatory processes, making them potential diagnostic and/or prognostic tools and also conceivable therapeutic targets for corresponding pathologies.

## Figures and Tables

**Figure 1 ijms-24-16962-f001:**
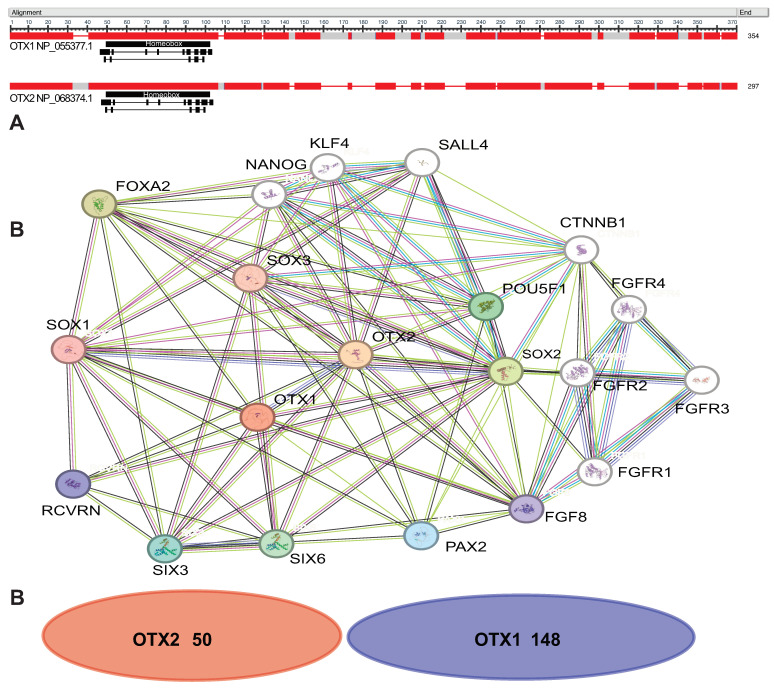
(**A**) Similarity alignment of OTX1 and OTX2 proteins. Red boxes represent high similarity protein sequence elaborated using COBALT Constraint-based Multiple Alignment Tool (NCBI). Black boxes represent the Homeobox domain common to the two transcription factors, small boxes the nucleotides involved in the DNA binding. (**B**) Interactors of OTX1 and OTX2 elaborated with STRING Version 12 software “https://string-db.org/ (accessed on 16 November 2023)”. Network nodes represent proteins, colored nodes represent query proteins (OTX1, 2) and first shell of interactors. The filled nodes indicate when a 3D structure is known or predicted. Edges represent protein–protein associations. The list of interactors has been generated using BioGRID version 4.4.225 “https://thebiogrid.org (accessed on 16 November 2023)”. Only interactors with physical, non-redundant high-throughput (HTP) evidence have been considered.

**Figure 2 ijms-24-16962-f002:**
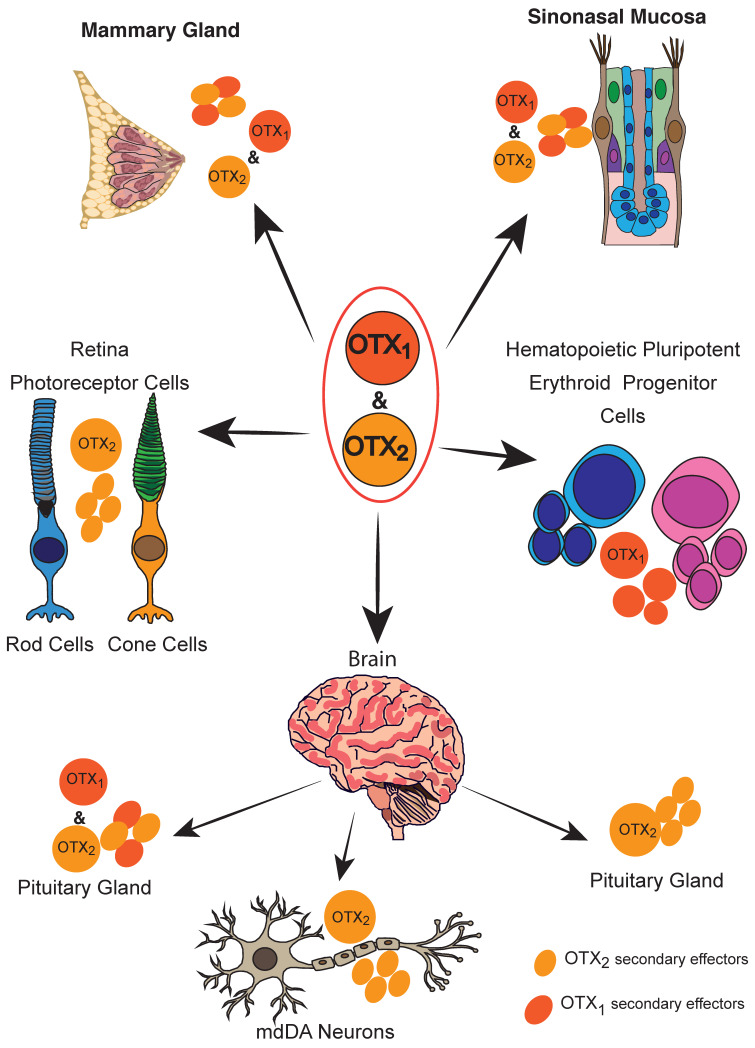
Schematic representation of OTX transcription factors involvement in development and differentiation in different organs.

**Figure 3 ijms-24-16962-f003:**
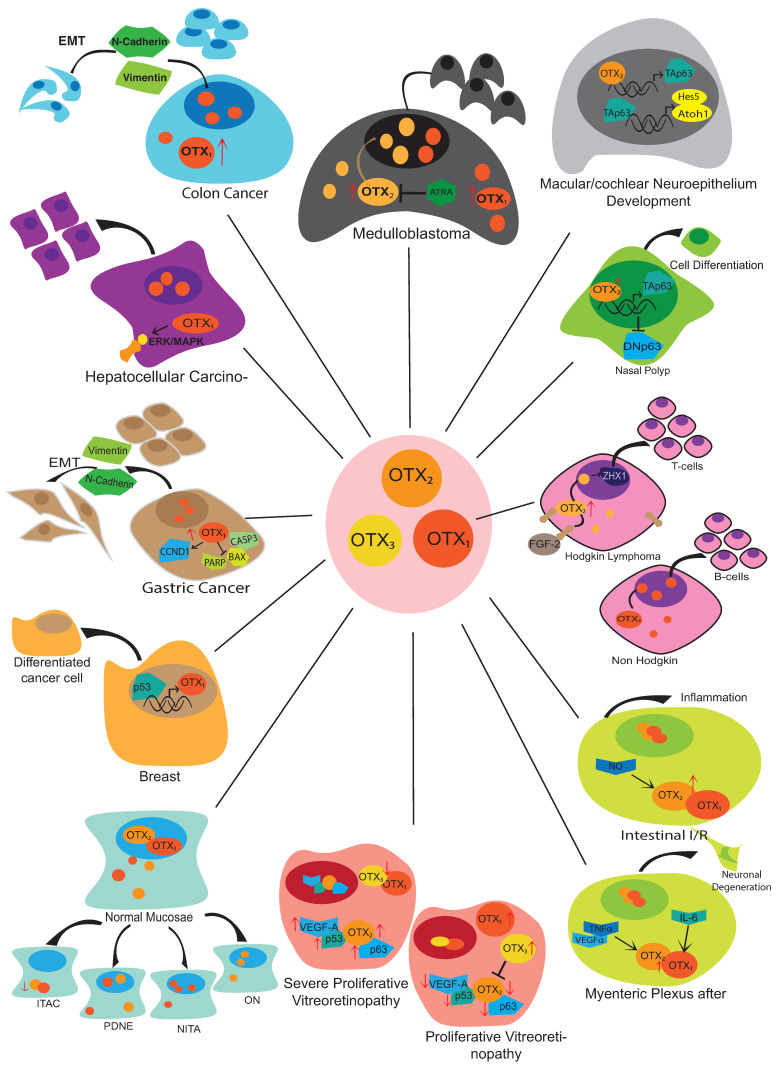
Schematic representation of OTX transcription factors’ involvement in pathological diseases. Red arrows indicate the effect of upregulation or downregulation of specific effectors. Interactions with other proteins are specifically indicated in different pathologic conditions. OTX1 and OTX2 often act on the same pathway and inside the same cell system, either forming dimers or complexes with other proteins.

**Figure 4 ijms-24-16962-f004:**
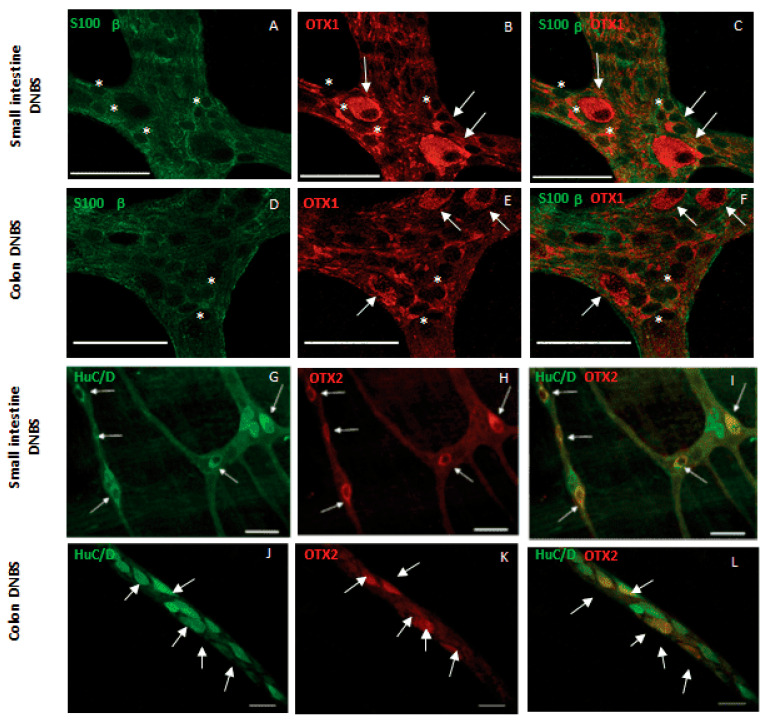
Confocal images showing immunohistochemical localization of OTX1 and OTX2 in longitudinal muscle myenteric plexus (LMMP) whole-mount preparations of the rat small intestine and colon after DNBS-induced colitis. (**A**–**F**) Co-localization of OTX1 with the glial marker S100 β. (**G**–**L**) Co-localization of OTX2 with the pan neuronal marker HuC/D. Arrows indicate neurons and asterisks indicate glial cells. With modifications from Bistoletti et al., 2020 [74]. Bar: 50 μm.

## Data Availability

No new data were created or analyzed in this study. Data sharing is not applicable to this article.

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
