# Peer review of "OTX Genes in Adult Tissues"

_ijms, 2023, doi:10.3390/ijms242316962_

Round 1

Reviewer 1 Report

Comments and Suggestions for Authors

2.11.23

 OTX genes in adult tissues

By: Alessandro Terrinoni1,Giovanni Micheloni, Vittoria Moretti, Sabrina Caporali, Sergio Bernardini, Marilena Minieri, Massimo Pieri, Cristina Giaroni, Francesco Acquati, Lucy Costantino, Fulvio Ferrara, Roberto Valli and Giovanni Porta

Submitted to Int. J. Mol. Sci

This is an interesting comprehensive review on the possible involvement of OTX genes in mature tissue, in both their proper function and in disease states, mostly in cancer.

I suggest the authors add a description of the genes and proteins (preferably in a figure) with all the important functional regions.  It may be also beneficial to add a graphical description of the associated transcription factors and other regulatory proteins that bind to/or are affected by these two important factors.

In the same context the authors should better explain the uncommon function of OTX genes as modulators of histone acetylation and methylation (lines 290-292).

Minor comments:

In figure 1 pituitary is misspelled

Line 49- the word: “Unfortunately” should be omitted.

Figure 2- please check the legend. The figure is not self-explanatory and the legend is not detailed enough, for example it is not clear if you are referring to breast, or breast cancer and what happens to the OTX genes during the different stages. Please explain “other complexes”.

Line 75- “sense organs”- should be “sensory organs”.

There are several places where the authors describe the “peculiar role”- please rephrase.

Line 124- please explain how “the French flag temporal model” is involved in these findings.

Line 149-151 should be rephrased.

Line 159- please explain is it protecting or damaging?

It seems that lines 278-288 contain conflicting findings, it should be better explained. 

Line 320- “developing” should be “development”.

Comments on the Quality of English Language

adequate, minor corrections are required

Reviewer 2 Report

Comments and Suggestions for Authors

Dear Authors,

this reviewer appreciated your effort in collecting the state of the art related to OTX1 and 2 function in healthy and pathological conditions. However, I have some suggestions t improve your MS.

Major comments:

1.Please, introduce a section regarding the role Otx genes have in embryonic and adult stem cells

2. Please, introduce a paragraph describing the involvement of OTX genes in diseases which are currently only mentioned in sections describing the role of OTX1 and 2 in adult tissues

3. Sections 3.2, 3.3, 4.11 are too short. Could  you extend these sub-paragraphs, giving more details about the mechanisms through which Otx genes act in these tissues/cancer?

4. The conclusion section is too short. Please comment further on the role Otx genes have in physiological and pathological conditions

Minor comments:

1. Line 34: please substitute "encoded" with "embedded" in chromosomes 2p13 and 14q21-11, respectively

2. Figure 1: indicate some of the tissue-specific otx secondary effectors3. Line 80: please re-phrase:  "Later, it is also expressed"

4. Line 140: what A10 stands for?

5. Lines 211-212: Please, re-phrase:  “We can thus suggest that OTX genes are implicated in neuronal degeneration during inflammatory states along the gastrointestinal tract, suggesting OTX genes as potential targets for the development of new therapeutic approaches”

6. Lines 288-289: which MB promoters? Which stem cells genes?

7. Line 340: please, substitute "also intriguing" with "Intriguingly"

8. Line 433: eliminate again

9. Lines 488-490: meaning is incomprehensible. Please, re-phrase

Comments on the Quality of English Language

English is almost fine throughout the text. However, sometimes, the meaning of the phrases is incomprehensible. Modifications to the text have been suggested.

Round 2

Reviewer 2 Report

Comments and Suggestions for Authors

Dear Authors,

I have no further requests. To this Reviewer the paper is acceptable in its present form